# Automatic Evaluation of the Lung Condition of COVID-19 Patients Using X-ray Images and Convolutional Neural Networks

**DOI:** 10.3390/jpm11010028

**Published:** 2021-01-04

**Authors:** Ivan Lorencin, Sandi Baressi Šegota, Nikola Anđelić, Anđela Blagojević, Tijana Šušteršić, Alen Protić, Miloš Arsenijević, Tomislav Ćabov, Nenad Filipović, Zlatan Car

**Affiliations:** 1Faculty of Engineering, University of Rijeka, Vukovarska 58, 51000 Rijeka, Croatia; ilorencin@riteh.hr (I.L.); sbaressisegota@riteh.hr (S.B.Š.); nandelic@riteh.hr (N.A.); 2Faculty of Engineering, University of Kragujevac, Sestre Janjić, 34000 Kragujevac, Serbia; andjela.blagojevic@kg.ac.rs (A.B.); tijanas@kg.ac.rs (T.Š.); fica@kg.ac.rs (N.F.); 3Bioengineering Research and Development Centre (BioIRC), Prvoslava Stojanovića 6, 34000 Kragujevac, Serbia; 4Clinical Hospital Centre, Rijeka, Krešimirova ul. 42, 51000 Rijeka, Croatia; alen.protic@uniri.hr; 5Faculty of Medicine, University of Rijeka, Ul. Braće Branchetta 20/1, 51000 Rijeka, Croatia; 6Clinical Centre Kragujevac, Zmaj Jovina 30, 34000 Kragujevac, Serbia; milos.arsenijevic83@gmail.com; 7Faculty of Medical Sciences, University of Kragujevac, Svetozara Markovića 69, 34000 Kragujevac, Serbia; 8Faculty of Dental Medicine, University of Rijeka, Krešimirova ul. 40, 51000 Rijeka, Croatia; tomislav.cabov@fdmri.uniri.hr

**Keywords:** AlexNet, convolutional neural network, COVID-19, ResNet, VGG-16

## Abstract

COVID-19 represents one of the greatest challenges in modern history. Its impact is most noticeable in the health care system, mostly due to the accelerated and increased influx of patients with a more severe clinical picture. These facts are increasing the pressure on health systems. For this reason, the aim is to automate the process of diagnosis and treatment. The research presented in this article conducted an examination of the possibility of classifying the clinical picture of a patient using X-ray images and convolutional neural networks. The research was conducted on the dataset of 185 images that consists of four classes. Due to a lower amount of images, a data augmentation procedure was performed. In order to define the CNN architecture with highest classification performances, multiple CNNs were designed. Results show that the best classification performances can be achieved if ResNet152 is used. This CNN has achieved $\bar{AUC_{macro}}$ and $\bar{AUC_{micro}}$ up to 0.94, suggesting the possibility of applying CNN to the classification of the clinical picture of COVID-19 patients using an X-ray image of the lungs. When higher layers are frozen during the training procedure, higher $\bar{AUC_{macro}}$ and $\bar{AUC_{micro}}$ values are achieved. If ResNet152 is utilized, $\bar{AUC_{macro}}$ and $\bar{AUC_{micro}}$ values up to 0.96 are achieved if all layers except the last 12 are frozen during the training procedure.

## 1. Introduction

The Coronavirus disease 2019 (COVID-19) caused by Severe Acute Respiratory Syndrome virus 2 (SARS-CoV-2) is a viral, respiratory lung disease [1]. The spread of COVID-19 has been rapid, and it has affected the daily lives of millions across the globe. The dangers it poses are well-known [2], with the most important ones being its relatively high severity and mortality rate [3] and the strain it exhibits on the healthcare systems of countries worldwide [4,5]. Another problematic characteristic that COVID-19 exhibits is a wide variation in severity across the patients, which can cause issues for healthcare workers who wish to determine an appropriate individual treatment plan [6]. Early determination of the severity of COVID-19 may be vital in securing the needed resources—such as planning the location for hospitalization of the patient or respiratory aids in case they may be necessary. There is a dire need for systems that will allow for the strain put on the resources of the healthcare systems, as well as healthcare workers, by allowing easier classification of patient case severity in early stages of hospitalization. Artificial intelligence (AI) techniques have already been proven to be a useful tool in the fight against COVID-19 [7,8], so the possibility exists of them being applied in this area as well. Existence of such algorithms may lower the strain on the potentially scarce resources, by allowing early planning and allocation. Additionally, they may provide decision support to overworked healthcare professionals.

Internet of Medical Things (IoMT) is a medical paradigm that allows for integration of modern technologies in the existing healthcare system [9]. The algorithms developed as a part of the presented research can be made available to health professionals using IoMT [10]. Models obtained using the described methodology can be integrated inside a pipeline system in which an X-ray image will automatically be processed using the developed models, and the predicted class of the patient whose image has been obtained will immediately be delivered to the medical professional examining the X-ray. Such automated diagnosis methods have already been applied in many studies, such as in histopathology [11], neurological disorders [12], urology [13], and retinology [14]. All the researchers agree that not only can such AI-based support systems provide an extremely precise diagnosis, but can also be integrated in automatic systems to provide assistance to medical experts in determining the correct diagnosis. The obtained models are suited for such an approach. While the training of the models is slow due to the backpropagation process, the classification (using forward propagation) is fast and computationally moderate [15,16], allowing for easy integration into existing in-hospital systems.

The machine learning diagnostic approach has been successfully applied to X-ray images a number of times in the past. For example, Lujan–Garcia et al. (2020) [17] demonstrated the application of CNNs for the detection of pneumonia using chest X-ray images using Xception CNN, which was pre-trained using a ImageNet dataset for initial values. The evaluation was performed using precision, recall, F1 score, and AUROC, with the achieved scores being 0.84, 0.99, 0.91, and 0.97, respectively. Kieu et al. (2020) [18] demonstrated the Multi-CNN approach to the detection of abnormalities on the chest X-ray images. The approach presented in the paper demonstrates the use of multiple CNNs to determine the class of the input image, with the hybrid system presented achieving an accuracy of 96%. Bullock et al. (2019) [19] presented XNet—a CNN solution designed for medical, X-ray image segmentation. The presented solution is suitable for small datasets, and achieves high scores (92% accuracy, F1 score of 0.92 and AUC of 0.98) on the used dataset. Takemiya et al. (2019) [20] demonstrated the use of R-CNNs (Region with Convolutional Neural Network) in the detection of pulmonary nodules from the images of chest X-ray images. The proposed method utilizes the Selective Search algorithm to determine the potential candidate regions of chest X-rays and applied the CNN to classify the selected regions into two classes—nodule opacities and non-nodule opacities. The presented approach achieved high classification accuracy. Another example is by Stirenko et al. (2018) [21], in which the authors applied the deep learning, CNN approach to the X-ray images of patients with tuberculosis. The CNN is applied to a small and non-balanced dataset with the goal of segmentation of chest X-ray images, allowing for classification of images with higher precision in comparison to non-segmented images. In combination with data augmentation techniques, the achieved results are better. Authors conclude that data augmentation and segmentation, combined with dataset stratification and removal of outliers, may provide better results in cases of small, poorly balanced datasets.

There was research that utilized transfer learning methodologies in order to recognize respiratory diseases from chest X-ray images. In [22], the authors proposed a transfer learning approach in order to recognize pneumonia from X-ray images. The proposed approach, based on utilization of ImageNet weights has resulted with high accuracy of pneumonia recognition (96.4%). Another transfer learning approach has been implemented for pneumonia detection in [23]. By utilizing such an approach, a highly accurate multi-class classification can be achieved, with accuracy ranging from 93.3% to 98%.

Wong et al. [24] (2020) noted that radiographic findings do indicate positivity in COVID-19 patients, the conclusions of which are further supported by Orsi et al. [25] (2020) and Cozzi et al. [26] (2020). Borghesi and Maroldi [27] (2020) defined a scoring system for X-ray COVID-19 monitoring, concluding that there is a definite possibility of determining the severity of the disease through the observation of X-ray images. Research has been done in the application of AI in the detection of COVID-19 in patients. Recently, classification of patients for preliminary diagnosis has been done from cough samples. Authors Imran et al. [28] (2020) applied and implemented this into an app, called AI4COVID-19.

Bragazzi et al. [29] (2020) demonstrated the possible uses of information and communication technologies, artificial intelligence, and big data in order to handle the large amount of data that may be generated by the ongoing pandemic. Further reviews and comparisons of mathematical modeling, artificial intelligence, and datasets for the prediction were done by multiple authors, such as Mohamadou et al. [30] (2020), Raza [31] (2020), and Adly et al. [32] 2020. All aforementioned authors concluded the possibility of application of AI in the current and possibly forthcoming pandemics. Most promise in AI applications being applied in this field has been shown in the field of epidemiological spread. Zheng et al. [33] (2020) applied a hybrid model for a 14-day period prediction, Hazarika et al. [34] (2020) applied wavelet-coupled random vector functional neural networks, while Car et al. [35] (2020) applied a multilayer perceptron neural network for the goal of regressing the epidemiology curve components. Ye et al. [36] (2020) demonstrated a α-Satellite, AI-driven system for risk assessment at a community level. Authors demonstrate the usability of such a system in combat against COVID-19, as a system that displays risk index and the number of cases among all larger locations across the United States. Authors in [37] have proposed a method for forecasting the impact of COVID-19 on stock prices. The approach based on stationary wavelet transform and bidirectional long short-term memory has shown high estimation performances.

Still, a large amount of work was also done in the image classification and detection of COVID-19 in patients. Wang et al. [38] (2020) demonstrated the use of high-complexity convolutional neural networks in the application of COVID-19 diagnosis. Their COVID-Net custom architecture reached high sensitivity scores (above 90%) in the detection of the COVID-19 in comparison to other infections and a normal lung state. Narin et al. [39] (2020) also demonstrated a high-quality solution using deep convolutional neural networks on X-ray images. Through the application of five different architectures (ResNet50, ResNet101, ResNet152, Inception V3, and Inception-ResNetV2) high scores were achieved (accuracy 95% or higher) by the authors. Ozturk et al. [40] (2020) developed a classification network for classifying the inflammation, named DarkCovidNet. DarkCovidNet reached an impressive score in binary classification at 98.08% in the case of binary classification, but a significantly lower score for 87.02% for multi-label classification. In the presented case, a multi-label classification was conducted with the aim of differentiating X-ray images of the lungs of healthy patients, patients with COVID-19, and patients with pneumonia. Abdulaal et al. [41] (2020) demonstrated the AI-based prognostic model of COVID-19, achieving accuracy levels of 86.25% and AUC ROC 90.12% for UK patients.

There have been studies proposing a transfer-learning approach to COVID-19 diagnosis from X-ray images of the chest. The study presented in [42] used pre-trained CNNs in order to automatically recognize COVID-19 infection. Such an approach has enabled high classification performances with an accuracy level of up to 99%. The research presented in [43] proposed a similar approach in order to differentiate pneumonia from COVID-19 infection. Transfer learning has enabled higher classification accuracy with utilization of simpler CNN architecture, such as VGG-16.

While a lot of work suggests that neural networks may be used for the detection of COVID-19 infection, there is an apparent lack of work that tests the possibility of finding the severity of COVID-19 through patients’ lung X-rays. Such an approach would allow for automatic detection and prediction of case severity, allowing healthcare professionals to determine the appropriate approach and to leverage available resources in the treatment of that individual patient. Development of an AI basis for such a novel system is the goal of this paper. From a literature overview, it can be noticed that all presented research has been based on a binary classification of X-ray images (infected/not infected) or differentiating COVID-19 infection and other respiratory diseases.

To summarize the novelty, this article, unlike the articles presented in the literature review, deals with a multi-class classification of X-ray images of positive COVID-19 patients with the aim of estimating the clinical picture. All the examples have used a large number of images (larger than 1000) in the training and testing processes of the neural network. While the number of COVID-19 patients is high, data collection, especially in countries with lower quality healthcare systems, may be problematic due to the strain exhibited by the coronavirus. Because of this, it is important to test the possibility of algorithm development combined with data augmentation operations, which is the secondary goal of the presented research.

According to presented facts and the literature overview, the following questions arise:Is it possible to utilize CNN in order to classify COVID-19 patients according to X-ray images of lungs?Which CNN architecture achieves the highest classification performance?Which are the best-performing configurations in regards to the solver, number of iterations, and batch size?How do transfer learning and layer freezing influence the performances of the best configurations?

## 2. Dataset Construction

In this section, a brief description of the used dataset will be provided, together with examples of each class. Furthermore, a data augmentation technique will be presented. At the end, divisions in the training, validation, and testing sets will be presented.

### 2.1. Dataset Description

The dataset used in this research was obtained from the Clinical Centre in Kragujevac [44] and consists of 185 X-ray images that represent the lungs of 21 patients diagnosed with COVID-19. The dataset consists of 7 female and 18 male patients, and age of patients in the form of mean ± standard deviation was 58.9±11.1 years. Images have been divided into four groups according to the clinical picture of the patient. Classification to a clinical picture was performed according to the clinical data that contained parameters such as:Clinical picture description;Physical examination;Laboratory examination; andX-ray finding.

According to the aforementioned division, images have been classified into classes:Mild clinical picture;Moderate clinical picture;Severe clinical picture; andCritical clinical picture.

An overview of image classes has been presented in Figure 1, where each class is represented with a X-ray image.

For the purposes of this research, X-ray images collected during treatment have been used to create a dataset. The dataset was created with respect to the clinical picture of the patient, where each X-ray image was classified to the appropriate class. Data distribution according to classes is presented in Figure 2.

### 2.2. Description of Data Augmentation Technique and Resulting Dataset

Due to a small amount of images in the dataset, a process called augmentation has been utilized in order to increase the classification performances [45]. The augmentation procedure was performed with the aim of artificially increasing the training dataset [46], while the testing dataset remained the same. This procedure is often used in fields such as bio-medicine, due to the fact that a large amount of bio-medical data is often unavailable [47]. In this particular case, a set of geometrical operations was utilized in order to increase the dataset. The aforementioned geometrical operations are:90 degree rotation around sagittal axis,180 degree rotation around sagittal axis,270 degree rotation around sagittal axis,180 degree rotation around longitudinal axis,180 degree rotation around longitudinal axis combined with 90 degree rotation around sagittal axis,180 degree rotation around longitudinal axis combined with 180 degree rotation around sagittal axis, and180 degree rotation around longitudinal axis combined with 270 degree rotation around sagittal axis.

In addition to the above list, brightness augmentation was also performed. All the images obtained by the geometrical transformations given above were further modified by multiplying all image pixel values with factors 0.8, 0.9, 1.1, and 1.2 in addition to the original brightness.

The 90-degree rotation presents an operation that rotates the original image (presented with Figure 3a) by 90 degrees in a clockwise direction around the sagittal axis, as presented in Figure 3b. Following the presented logic, rotations for 180 and 270 degrees were performed as well, as presented in Figure 3c,d. Images rotated by 90 and 270 degrees were resealed in order to have the same dimensions as the original image. Image generation by 180-degree rotation around the longitudinal axis was performed in such a way that the new image represented a mirrored projection of the original image, as presented in Figure 3e. The mirrored image was rotated around the sagittal axis, forming three new variations, as presented in Figure 3f–h. As the final approach to image augmentation, a process of multiplication of all image pixels with a certain factor is proposed. In this case, four different factors (0.8, 0.8, 1.1, and 1.2) were used. The described transformations have been presented on an original image with Figure 3i–l. It is important to notice that such transformations were applied on an augmented set that was created by using all described geometrical transformations. By using such an approach, the new augmented dataset was four times larger than the dataset created by using just geometrical transformations.

Only geometrical transformations and multiplication of all image pixels with a certain factor were used for data augmentation in order to keep the entire data of the image. Other techniques, such as scaling, could remove parts of the image, so they were considered inappropriate due to the nature of the problem, which observes the entire image as it is delivered from the hospital X-ray system.

By using the augmentation process described in the previous paragraphs, a new augmented dataset of 5400 images was constructed. The class distribution of the new set is presented in Figure 4a. It is important to notice that for the creation of the augmented dataset, only images contained in the original training set were used. In other words, images used for classifier testing were not used for the creation of the augmented set. According to the presented fact, the training set of 881 images was divided into training and validation sets in a 75:25 manner, as a ratio common in machine-learning practice. The presented sets were used for the training of CNNs, while the original testing set was used for the evaluation of their classification performances. The above-described dataset division is presented in Figure 4b.

## 3. Description of Used Convolutional Neural Networks

In this subsection, an overview of CNN-based methods for image classification will be presented. The CNNs used in this research are, in fact, standard CNN architectures widely used for solving various computer vision and image recognition problems [48]. Such an algorithm, alongside its variations, is widely used for various tasks of medical image recognition [49]. For the case of this research, four different CNN architectures were used, and they are:AlexNet,VGG-16, andResNet.

All of the above-listed CNN architectures have predefined layers and activation functions, while other hyper-parameters, such as batch size, solver, and number of epochs could be varied. The above-listed architectures were chosen due to the history of their high classification performances in similar problems. It has been shown that ResNet architectures have achieved high classification performances when used for multi-class classification of X-ray chest images [50]. Furthermore, ResNet architectures were used in various tasks of medical data classification ranging from tumor classification [51,52], trough recognition of respiratory diseases [53], to fracture diagnosis [54,55].

Extensive searches for the optimal solution through the hyper-parameter space can also be called the grid-search procedure. Variations of hyper-parameters used during the grid-search procedure for CNN-based models are presented in Table 1.

In order to determine the influence of overfitting, the number of epochs were varied with the aim of determining the number with the highest performances on the test dataset. With respect to theoretical knowledge, it can be defined that when training with a large number of epochs, the model is often over-fitted. For this reason, it is necessary to find the optimal number of training epochs [15]. Solvers used in this research were selected due to their performance on multiple multi-label datasets [59]. In the following paragraphs, a brief description and mathematical models will be provided for each solver.

Adam Solver

The Adam optimization algorithm represents one of the most-used algorithms for tasks of image recognition and computer vision. By using the Adam optimizer, weights are updated by following [56]:(1)$$w_{t}^{i}=w_{t-1}^{i}-\frac{\eta }{\sqrt{\hat{v_{t}}}+\epsilon }\hat{m_{t}},$$
where $\hat{m_{t}}$ is defined as:(2)$$\hat{m_{t}}=\frac{m_{t}}{1-\beta_{1}^{t}}$$
and $\hat{v_{t}}$ is defined as:(3)$$\hat{v_{t}}=\frac{v_{t}}{1-\beta_{2}^{t}}.$$

$m_{t}$ is defined as a running average of the gradients, and it can be described with:(4)$$m_{t}=\beta_{1}m_{t-1}+\left(1-\beta_{1}\right)G.$$

Furthermore, $v_{i}$ is defined as the running average of squared gradients, or:(5)$$v_{t}=\beta_{2}v_{t-1}+\left(1-\beta_{2}\right)G^{2}.$$

*G* can be defined with:(6)$$G=\nabla_{w}C\left(w_{t}\right),$$
where C(arg) represents a cost function. Parameters of the Adam solver used in this research are presented in Table 2.

AdaMax Solver

AdaMax solver follows the logic similar to the Adam solver—in this case, the weights update was performed as [58]:(7)$$w_{t}^{i}=w_{t-1}^{i}-\frac{\eta }{v_{t}+\epsilon }\hat{m_{t}},$$
where $\hat{m_{t}}$ is defined as:(8)$$\hat{m_{t}}=\frac{m_{t}}{1-\beta_{1}^{t}}.$$

Furthermore, $v_{t}$ can be defined as:(9)$$v_{t}=max(\beta_{2}v_{t-1},|G_{t}\left|\right),$$
and $m_{t}$ is defined as:(10)$$m_{t}=\beta_{1}m_{t-1}+\left(1-\beta_{1}\right)G.$$

As it is in the case of the Adam solver, parameters used in this research are presented in Table 2.

Nadam Solver

The third optimizer used in this research in Nadam. As with the AdaMax algorithm, Nadam is also based on Adam. Weights in these case updates are as [58]:(11)$$w_{t}^{i}=w_{t-1}^{i}-\frac{\eta }{\sqrt{v_{t}}+\epsilon }\tilde{m_{t}},$$
where $\tilde{m_{t}}$ is defined with:(12)$$\tilde{m_{t}}=\beta_{1}^{t+1}\hat{m_{t}}+\left(1-\beta_{1}^{t}\right)\hat{g_{t}}.$$

$\hat{m_{t}}$ and $\hat{g_{t}}$ are defined as:(13)$$\hat{m_{t}}=\frac{m_{t}}{\prod_{i=1}^{t}\beta_{1}}$$
and
(14)$$\hat{g_{t}}=\frac{g_{t}}{\prod_{i=1}^{t}\beta_{1}}.$$

As it is in the case of the Adam and AdaMax optimizers, the parameters of the Nadam solver are presented in Table 2.

The presented parameters will be used for training the CNNs, and the classification performances of all trained models will be evaluated by using the testing data set. In the following paragraphs, a brief overview of the used CNN architectures will be presented.

### 3.1. AlexNet

AlexNet represents one of the classical CNN architectures that are used for various tasks of image recognition and computer vision. This architecture is one of the first CNNs that are based on deeper configuration [60]. AlexNet won the ImageNet competition in 2012. The success of such a deep architecture has introduced a trend for designing even deeper CNNs that can be noticed today [61]. AlexNet is based on a configuration of nine layers, where the first five layers are convolutional and pooling layers, and the last four are fully connected layers [62]. The detailed description of AlexNet architecture in provided in Table 3.

### 3.2. VGG-16

The described trend of deeper CNN configuration resulted in improvements of the original AlexNet architecture. One of such architectures is VGG-16, presented in the following year. VGG-16 represents a deeper version of AlexNet, where the nine-layer configuration is replaced with a 16-layer configuration, from which the name is derived [63]. A main advantage of VGG-16 is the introduction of smaller kernels in convolutional layers, in comparison with AlexNet [64]. The detailed description of VGG-16 layers is provided in Table 4.

### 3.3. ResNet

According to the presented networks, the trend of designing deeper networks can be noticed [65]. This approach can be utilized to a certain level, due to the vanishing gradient problem [66]. It can be noticed that deeper configurations will have no significant improvements in terms of classification performances. Furthermore, in some cases, deeper CNNs can show lower classification performances than CNNs designed with a smaller number of layers. For these reasons, an approach based on residual blocks is proposed. The residual block represents a variation of a CNN layer, where a layer is bypassed with an identity connection [67]. The block scheme of such an approach is presented in Figure 5.

By using the presented residual approach, significantly deeper networks could be used without the vanishing gradient problem. This characteristic is a consequence of identity bypass utilization because identity layers do not influence the CNN training procedure [68]. For these reasons, deeper CNNs designed with a residual block will not produce the higher error in comparison with shallower architectures. In other words, by stacking residual layers, significantly deeper architectures could be designed. For the case of this research, three different architectures based on the residual block will be used, and these are: ResNet50 [69], ResNet101 [70], and ResNet152 [71]. The aforementioned architectures are pre-defined ResNet architectures that are mainly used for image recognition and computer vision problems which require deeper CNN configurations.

## 4. Research Methodology

As presented in the previous sections, this research is based on a comparison of multiple methods of image recognition that will be used in order to estimate the severity of COVID-19 symptoms according to X-ray images of patients’ lungs. All methods have been compared and evaluated from a standpoint of classification performances. In this case, $\bar{AUC_{micro}}$ and $\bar{AUC_{macro}}$ are used.

### 4.1. Description of $\bar{AUC_{micro}}$ and $\bar{AUC_{macro}}$

Image classifiers are evaluated using standard classification measures, such as the Area under the ROC curve (AUC). Such an approach is based on construction of the ROC curve by using a false-positive rate (FPR) and true-positive rate (TPR). TPR can be described as a ratio between the number of correct classifications in one class ($A_{C}$) and the sum of total members of that class. Such a number includes the number of correct classifications and the number of incorrect classifications ($A_{I}$). The aforementioned ratio can be defined as:(15)$$TPR=\frac{A_{C}}{A_{C}+A_{I}}.$$

On the other hand, FPR can be defined as a ratio of the number of incorrect classifications in the first class ($B_{I}$) and the total number of members of the second class ($B_{I}+B_{C}$). The aforementioned ratio can be written as:(16)$$FPR=\frac{B_{I}}{B_{I}+B_{C}}.$$

By using TPR and FPR, the ROC curve can be constructed and the AUC value can be determined. The challenge, in this case, lies in the fact that this measure is designed to evaluate the binary classifier. In the case of this research, the classification is performed in four classes. For this reason, a standard ROC-AUC procedure must be adapted to evaluate multi-class classification performances [49]. This approach is achieved by using $\bar{AUC_{micro}}$ and $\bar{AUC_{macro}}$ measures.

#### 4.1.1. $\bar{AUC_{micro}}$

The definition of $\bar{AUC_{micro}}$ is based on the calculation of $TPR_{micro}$ and $FPR_{micro}$. $TPR_{micro}$ can be calculated as a ratio between the number of correct classifications and the total number of samples. This relation can be written as:(17)$$\bar{TPR_{micro}}=\frac{A_{C}+B_{C}+C_{C}+D_{C}}{N},$$
where $A_{C}$ represents the number of correct classifications in the class *A*, $B_{C}$ the number of correct classifications in the class *B*, $C_{C}$ the number of correct classifications in the class *C*, and $D_{C}$ the number of correct classifications in the class *D*, where *N* represents the total number of samples. Following a similar methodology, $FPR_{micro}$ can be calculated as a ratio between the total number of incorrect classifications and the total number of samples. According to the above-stated notation, this ratio can be written as:(18)$$FPR=\frac{N-(A_{C}+B_{C}+C_{C}+D_{C})}{N}.$$

When the described $\bar{TPR_{micro}}$ and $\bar{FPT_{micro}}$ are used for ROC curve construction, the area underneath is called $AUC_{micro}$. This area represents a discrete micro-average value for the evaluation of multi-class classifier performances.

#### 4.1.2. $\bar{AUC_{macro}}$

Similar to $\bar{AUC_{micro}}$, $\bar{AUC_{macro}}$ can be used for performance evaluation of a multi-class classifier. In this case, average TPR is calculated as an average of TPR values that represent individual classes. For example, the TPR value for the class A can be calculated as a ratio between the number of correct classifications in the class A ($A_{C}$) and the total number of class A members ($N_{A}$). Such a ratio can be written as:(19)$$TPR_{A}=\frac{A_{C}}{N_{A}}.$$

When the presented formalism is applied to all classes, $\bar{TPR_{macro}}$ can be calculated as follows:(20)$$TPR_{macro}=\frac{1}{M}\sum_{n=1}^{M}TRP_{n},$$
where *M* represents the total number of classes. Following the presented procedure, $\bar{FPR_{macro}}$ can be calculated as an average of individual FPR values:(21)$$\bar{FPR_{macro}}=\frac{1}{M}\sum_{n=1}^{M}FRP_{n},$$
where the individual value can be calculated as a ratio between the number of incorrectly classified images as members of a particular class and the total number of images that are members of the same class. By using these measures, $\bar{AUC_{macro}}$ can be calculated.

### 4.2. Overfitting Issue

Due to the large CNN models used in this research, it is necessary to include steps to overcome overfitting. Over-fitted CNN shows high classification performances on the training dataset, while the performances on the testing dataset are quite poor. In order to prevent overfitting, some steps must be taken. According to [58], there are several mechanisms used to overcome overfitting in image classifiers. The mechanisms used in this research are:Image augmentation; andEarly stopping.

Image augmentation, as one of the key techniques for handling the overfitting issue, was addressed earlier in the article. In order to perform early stopping, an analysis regarding the change of $\bar{AUC_{micro}}$ and $\bar{AUC_{micro}}$ over the number of epochs was performed. Data obtained with this analysis will be used to determine the optimal number of training epochs for each CNN architecture. By using this approach, selected networks will be trained for the number of epochs which will allow for full training, while avoiding overfitting.

### 4.3. Freezing Layers

In order to increase classification performances of proposed networks, an approach of layers freezing during training procedure will be used. Such an approach will be performed on the CNN configurations that have already achieved the highest performances. The procedure of freezing layers will be performed in an iterative manner from the bottom of the network towards the higher layers until the maximal classification performances are achieved. Such a procedure is selected in order to fine-tune only specific layers of a CNN architecture, pre-trained with ImageNet, while other layers remain frozen during a training procedure. By using such an approach, issues regarding unscientific datasets are overcome to some extent. An example of a freezing layers methodology is presented on a ResNet architecture in Figure 6, where the first, second, third, and half of the fourth block are frozen during training, while other layers remain unfrozen.

### 4.4. Results Representation

In order to define the network that achieves the best classification performances, maximal $\bar{AUC_{micro}}$ and $\bar{AUC_{macro}}$ achieved with all networks will be compared. As a first step, the influence of the number of epochs and the batch size on maximal $\bar{AUC_{micro}}$ and $\bar{AUC_{macro}}$ will be examined. Furthermore, the configuration that produces the highest result will be presented for all CNNs. As a final step, all maximal $\bar{AUC_{micro}}$ and $\bar{AUC_{macro}}$ values achieved with each CNN will be compared in order to determine the architecture with the highest classification performances. A schematic representation of the research methodology is presented in Figure 7.

## 5. Results and Discussion

In this section, an overview of results achieved with each of the proposed CNN architectures will be presented. For each aforementioned architecture, diagrams that describe the change of maximal $\bar{AUC_{micro}}$, $\bar{AUC_{macro}}$ value in dependence of number of epochs and batch size will be provided. At the and of the section, a comparison of the achieved results will be presented and discussed.

### 5.1. Results Achieved with AlexNet

As the first of the results achieved with AlexNet architecture, the change of $\bar{AUC_{macro}}$ over the number of training epochs is presented in Figure 8. When the results are compared, it can be noticed that $\bar{AUC_{macro}}$ achieved its maximum at 50 and 75 training epochs, regardless of the solver utilized. Furthermore, it can be noticed that in the higher number of epochs, a significant fall of $\bar{AUC_{macro}}$ value occurs in the case of all solvers. Such a fall in classification performances could be recognized as a consequence of overfitting.

The change of $\bar{AUC_{micro}}$ is presented in Figure 9, where a similar trend as in the case of $\bar{AUC_{micro}}$ can be noticed. In this case, maximal performances are also achieved with 50 and 75 consecutive training epochs. Furthermore, it can be noticed that $\bar{AUC_{micro}}$ values are, at the same point, slightly higher than $\bar{AUC_{macro}}$. The trend of overfitting on a larger number of epochs is also noticeable.

When the influence of batch size on $\bar{AUC_{macro}}$ is observed, it can be noticed that there is no configuration that achieves a $\bar{AUC_{macro}}$ value higher than 0.8. This property is in correlation with the case described with Figure 8. It is interesting to notice a significant fall of $\bar{AUC_{macro}}$ values in the case when batches of size 16 are utilized. For this case, $\bar{AUC_{macro}}$ is set around a value of 0.7. This characteristic can be noticed for all three solvers utilized, as presented in Figure 10. Presented results are in correlation with previous knowledge regarding a regularizing effect of smaller batch sizes [72]. Such an approach has enabled overcoming of the overfitting issue.

As the final evaluation of AlexNet’s classification performances, the influence of batch size on $\bar{AUC_{micro}}$ will be observed. Similar to the case presented in Figure 9, an $\bar{AUC_{micro}}$ slightly higher than 0.8 was achieved if batches of four and eight were used. In the case of a batch size of 16, significantly lower $\bar{AUC_{micro}}$ around 0.7 was achieved. The described property can be noticed regardless of the solver utilized, as presented in Figure 11. These results are in correlation with results presented in the case of $\bar{AUC_{macro}}$.

### 5.2. Results Achieved with VGG-16

The results, similar to the results achieved with AlexNet, are achieved with VGG-16, as presented in Figure 12. Maximal $\bar{AUC_{macro}}$ values are achieved when the network is trained for 50 and 75 epochs. On the other hand, a significant decrease of $\bar{AUC_{macro}}$ can be noticed when the network is trained for a larger number of epochs. These lower results are a consequence of overfitting on a larger number of epochs.

Similar behavior of $\bar{AUC_{micro}}$ is presented in Figure 13, where maximal performances could be noticed when the network was trained for 50 or 75 consecutive epochs. When the network was trained for a higher number of epochs, a significant fall of $\bar{AUC_{micro}}$ could be noticed. Such a result is a consequence of overfitting on the larger number of epochs.

The influence of batch size on $\bar{AUC_{micro}}$ for the case of VGG-16 is presented in Figure 14. In the case of $\bar{AUC_{macro}}$, the maximal values are achieved when batches of four and eight are utilized, regardless of solver utilized. For the case of a batch size of 16, classification performances, with a value of 0.5, fall into the domain of the coin-flip classification. Such a result can be attributed to the regularization character of smaller batch-sizes and overfitting of larger batch-sizes.

When $\bar{AUC_{micro}}$ is measured, it can be noticed that the highest values are achieved when batches of four and eight are used. In the case when larger batches of 16 are used, $\bar{AUC_{micro}}$ value is positioned around a value of 0.7, as presented in Figure 15. In this case, a gap between $\bar{AUC_{micro}}$ and $\bar{AUC_{macro}}$ can also be noticed.

### 5.3. Results Achieved with ResNet Architectures

In the following sub-section, an overview of results achieved by using ResNet architectures will be presented. All results will be presented and described in a similar manner as in the case of AlexNet and VGG-16.

#### 5.3.1. Results Achieved with ResNet50

The change of $\bar{AUC_{macro}}$ over the number of epochs is presented in Figure 16. From the presented results, it can be noticed that the maximal $\bar{AUC_{macro}}$ values are achieved when the network is trained for 100 epochs. This characteristic can be noticed only for the case of the Adam and Adamax solvers, while for the case of the Nadam solver, the maximal $\bar{AUC_{macro}}$ is achieved when the network is trained for 50 consecutive epochs. If the CNN is trained for a larger number of epochs, a significant drop of $\bar{AUC_{macro}}$ can be noticed. Such a result is pointing towards the fact that the overfitting issue occurs if ResNet50 is trained for a larger number of epochs.

Furthermore, when Figure 16 and Figure 17 are observed, a similar trend can be noticed for the case of $\bar{AUC_{micro}}$. A significant drop of $\bar{AUC_{micro}}$ occurs if ResNet50 is trained for a higher number of consecutive epochs, while the $\bar{AUC_{micro}}$ value tops when the network is trained for 75 epochs with an Adam solver or 100 epochs for the AdaMax and Nadam solvers. The lower performances at the higher number of epochs are pointing towards overfitting.

When the influence of batch size on $\bar{AUC_{micro}}$ and $\bar{AUC_{macro}}$ is examined, the results presented in Figure 18 and Figure 19 are achieved. It can be noticed that the highest results are achieved when larger batches of 16 are used. In this case, the maximal $\bar{AUC_{macro}}$ will go up to 0.9 only if the AdaMax solver is utilized. In the case of smaller batches, the $\bar{AUC_{macro}}$ values between 0.7 and 0.8 are achieved, regardless of solver utilized.

Similar conclusions could be drawn when $\bar{AUC_{micro}}$ values are compared. In this case, the only significant difference is a significant underperformance of networks trained with the Adam solver by using smaller batches, as presented in Figure 19.

#### 5.3.2. Results Achieved with ResNet101

The change of $\bar{AUC_{macro}}$ over the number of epochs achieved with ResNet101 is presented in Figure 20. From the presented results, it can be noticed that the highest performances are achieved when the CNN is trained for 150 epochs. Such a property can be noticed for all solvers, with an exception of Nadam, which has achieved similar results at 50 epochs. Furthermore, the significant drop of $\bar{AUC_{macro}}$ value can be noticed at a higher number of epochs. Such a fall can be attributed to overfitting.

A similar trend is presented in Figure 21, where the change of $\bar{AUC_{micro}}$ value over the number of epochs is presented. As it is in the case of $\bar{AUC_{macro}}$, the maximal classification performances are achieved when CNNs are trained for 100, 125, and 150 consecutive epochs. Due to overfitting, significantly lower $\bar{AUC_{micro}}$ values are achieved when CNNs are trained for 175 and 200 epochs. The drop of $\bar{AUC_{micro}}$ value is noticeably deeper than in case of $\bar{AUC_{macro}}$.

The influence of batch size on $\bar{AUC_{macro}}$ is presented in Figure 22. When the results are observed, it can be noticed that the highest classification performances are achieved when larger data batches are used during training of ResNet101. These characteristics are noticed only for the case of the AdaMax and Adam solvers. On the other hand, when the Adam solver is used, no significant difference of $\bar{AUC_{macro}}$ is achieved when batches of 8 and 16 are used during training.

A similar conclusion could be reached if classification performances are evaluated by using $\bar{AUC_{micro}}$. It can be noticed that the highest $\bar{AUC_{macro}}$ values will be achieved if the network is trained by using larger batches, as presented in Figure 23.

#### 5.3.3. Results Achieved with ResNet152

The last CNN used in this research is ResNet152. The change of $\bar{AUC_{macro}}$ over number of epochs is presented in Figure 24. From the presented result, it can be noticed that the highest $\bar{AUC_{macro}}$ value is achieved when ResNet152 is trained for 125 epochs. Such a property can be noticed regardless of the solver utilized. An exception can be noticed in the case of the AdaMax solver. In this case, $\bar{AUC_{macro}}$ values over 0.9 are also achieved when the network is trained for 75 epochs. Furthermore, an influence of overfitting can be noticed when the network is trained for a larger number of epochs. Due to this property, it is important to train the network for a lower number of consecutive epochs in order to prevent overfitting and, consequently, lower classification performances.

Similar results are achieved when the change of $\bar{AUC_{micro}}$ over different number of epochs is observed, as presented in Figure 25. In this case, the highest $\bar{AUC_{micro}}$ values are achieved when the network is trained for 100 and 125 consecutive epochs. It is important to notice that a significant fall of $\bar{AUC_{micro}}$ occurs when the CNN is trained for 175 and 200 consecutive epochs. Such a trend can be noticed regardless of solver utilized, and it points toward an occurrence of overfitting. Due to these results, it can be concluded that it is necessary to avoid training for such a large number of epochs in order to prevent overfitting and to achieve higher classification performances.

When the influence of batch size on $\bar{AUC_{macro}}$ is observed, it can be noticed that by using a larger batch size of 16, higher $\bar{AUC_{macro}}$ will be achieved. The significantly lower $\bar{AUC_{macro}}$ values are achieved when ResNet152 is trained by using smaller batches of four. This property can be noticed for the case of all three solvers, as presented in Figure 26.

Similar results can be noticed when the influence of batch size on $\bar{AUC_{micro}}$ is observed. The only significant difference lies in the fact that for a batch size of four, somewhat higher values are achieved, as presented in Figure 27. Regardless of higher value, $\bar{AUC_{micro}}$, in this case, is still too low to be taken into consideration for practical application.

### 5.4. Comparison of Achieved Results

When the result achieved with all CNN architectures is compared, it can be noticed that in the case of AlexNet and VGG-16, the highest $\bar{AUC_{macro}}$ values are achieved if networks are trained by using smaller batches for a lower number of epochs. On the other hand, ResNet architectures show better performances when trained by using larger batches for a higher number of consecutive epochs. Configurations that have achieved the largest $\bar{AUC_{macro}}$ values are presented in Table 5.

Similarly to the above-presented architectures, the highest $\bar{AUC_{micro}}$ values are achieved when AlexNet and VGG-16 are trained by using smaller batches for a lower number of consecutive epochs. Furthermore, for the case of ResNet architectures, the highest $\bar{AUC_{micro}}$ values are achieved when CNNs are trained by using larger batches for a larger number of epochs. The described configurations are presented in Table 6.

Finally, when the highest $\bar{AUC_{macro}}$ and $\bar{AUC_{micro}}$ achieved with each CNN architecture are compared, it can be noticed that ResNet architectures are achieving dominantly higher classification performances. On the other hand, it can be noticed that by using deeper ResNet architectures, a rising trend of $\bar{AUC_{macro}}$ and $\bar{AUC_{micro}}$ is present, as presented in Figure 28. The achieved results are pointing to the conclusion that by using ResNet152 architecture, the highest $\bar{AUC_{macro}}$ and $\bar{AUC_{micro}}$ values of 0.93 and 0.94 are achieved. Given the results achieved, the possibility of using CNN for automatic classification of patients with COVID-19 with respect to lung status should be considered.

Furthermore, when layer freezing is considered, it can be noticed that by freezing higher layers of CNNs during the training procedure, higher classification performances are achieved. The distribution of frozen and unfrozen layers is presented in Table 7 for each CNN architecture utilized.

When comparing the achieved classification performances with classification performances in the case when all layers are fine-tuned, it can be noticed that in the case of freezing layers, slightly higher performances are achieved with each of the proposed CNN architectures, as presented in Figure 29.

Furthermore, it can be noticed how the order of the architectures from the one with the best classification performance to the one with the worst is the same as in the previous case.

When the results achieved with transfer learning are compared to the results achieved on similar problems, it can be noticed that higher classification performances are achieved if transfer learning is utilized. Such a correlation can be noticed when the achieved results are compared with the results of research dealing with both COVID-19 [42,43] and other respiratory issues [22,23]. These results are pointing towards the utilization of transfer learning in order to increase the accuracy of evaluation of the clinical picture of COVID-19 patients from X-ray lung images.

These results show that ResNet152, in combination with transfer learning, is the network that achieves the best results in the case of evaluation of the clinical picture of COVID-19 patients using X-ray lung images.

## 6. Conclusions

The results achieved with this research are pointing towards the conclusion that CNN-based architectures could be used in estimation of the clinical picture of a COVID-19 patient according to the X-ray lung images. It is important to notice that deep CNN architectures have the tendency to overfit when they are trained with a higher number of consecutive epochs. Due to this property, it is concluded that steps such as early stopping and image augmentation must be used in order to prevent overfitting. According to the presented results and stated research hypothesis, the following conclusions could be drawn:It is possible to utilize CNN for automatic classification of COVID-19 patients according to X-ray lung images;The best results are achieved if ResNet152 architecture is utilized;The best results are achieved if the aforementioned architecture is trained by using larger batches of data for an intermediate number of consecutive epochs by using Nadam solver; andIt can be noticed that by utilization of transfer learning and freezing layers, higher classification performances are achieved.

Due to the presented results and conclusions, a possibility for utilization of such an algorithm in battle against COVID-19 and its application in clinical practice should be taken into account. The main limitation of this research was the small amount of X-ray images, which could be overcome, to some extent, by augmentation techniques, and another limitation was class imbalance. Regardless of the presented limitations, the presented approach has shown promising results which point to further research on a larger and more balanced data set.

## Figures and Tables

**Figure 1 jpm-11-00028-f001:**
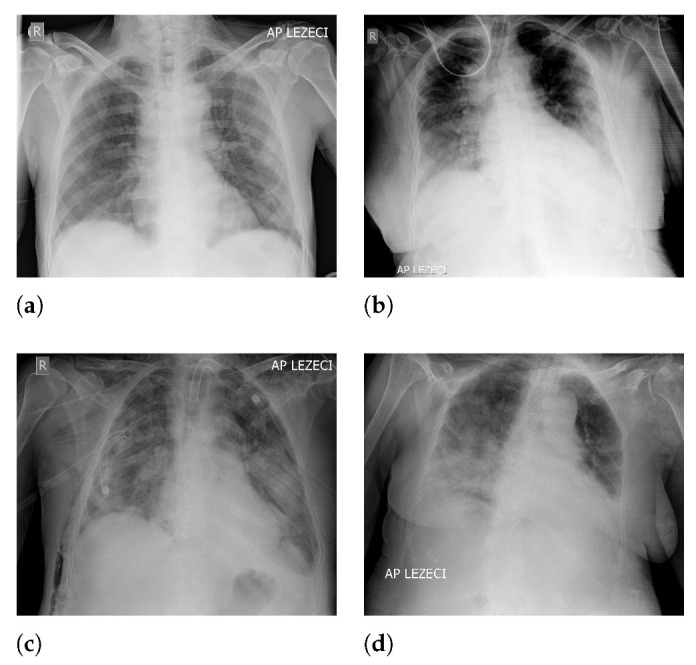
Examples of X-ray images contained in the dataset: (**a**) A mild clinical picture; (**b**) moderate clinical picture; (**c**) severe clinical picture; and (**d**) critical clinical picture.

**Figure 2 jpm-11-00028-f002:**
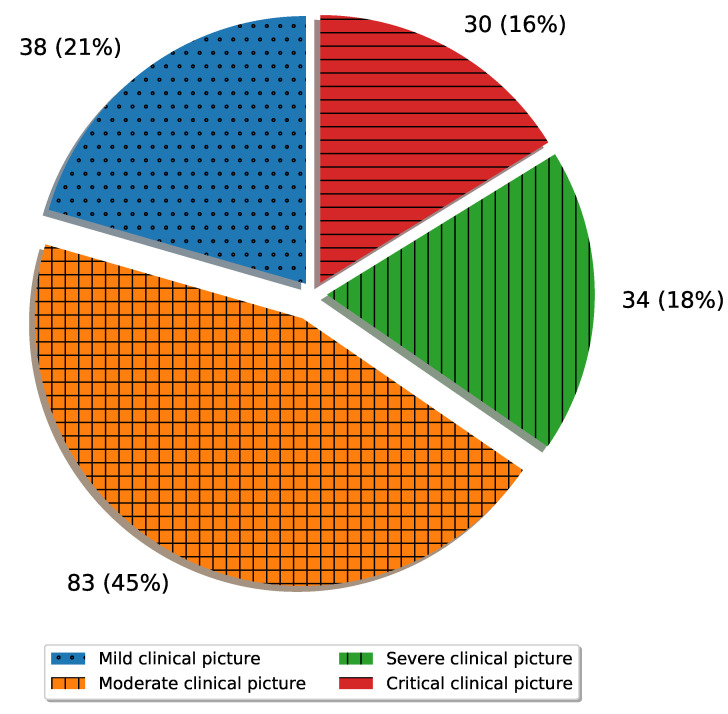
Overview of dataset distribution.

**Figure 3 jpm-11-00028-f003:**
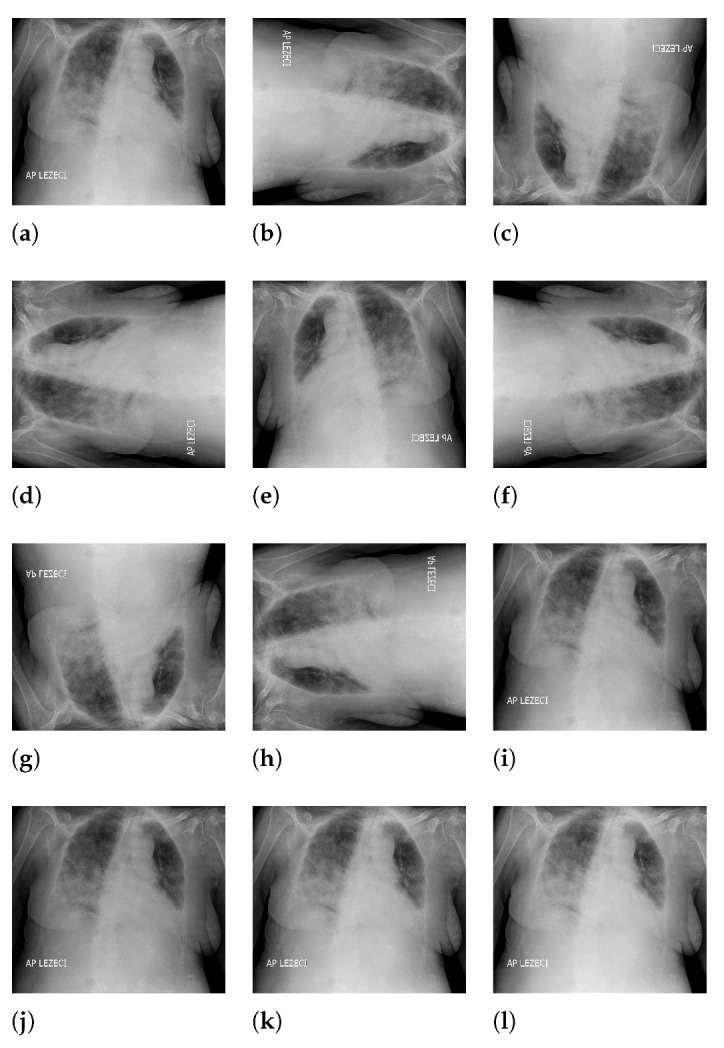
Overview of image augmentation procedure ((**a**): original image; (**b**): image rotated for 90 degrees around sagittal axis; (**c**): image rotated for 180 degrees around sagittal axis; (**d**): image rotated for 270 degrees around sagittal axis; (**e**): image rotated for 180 degree around longitudal axis; (**f**): image rotated for 180 degree around longitudal axis and rotated for 180 degree around sagittal axis; (**g**): image rotated for 180 degree around longitudal axis and rotated for 180 degree around sagittal axis; (**h**): image rotated for 180 degree around longitudal axis and rotated for 270 degree around sagittal axis; (**i**): image with pixels multiplied by a factor 0.8; (**j**): image with pixels multiplied by a factor 0.9; (**k**): image with pixels multiplied by a factor 1.1; (**l**): image with pixels multiplied by a factor 1.2).

**Figure 4 jpm-11-00028-f004:**
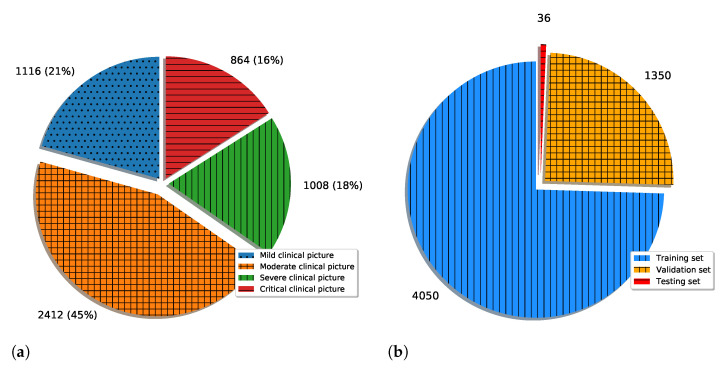
Representation of augmented dataset ((**a**): class distribution; (**b**): training-validation-testing division).

**Figure 5 jpm-11-00028-f005:**
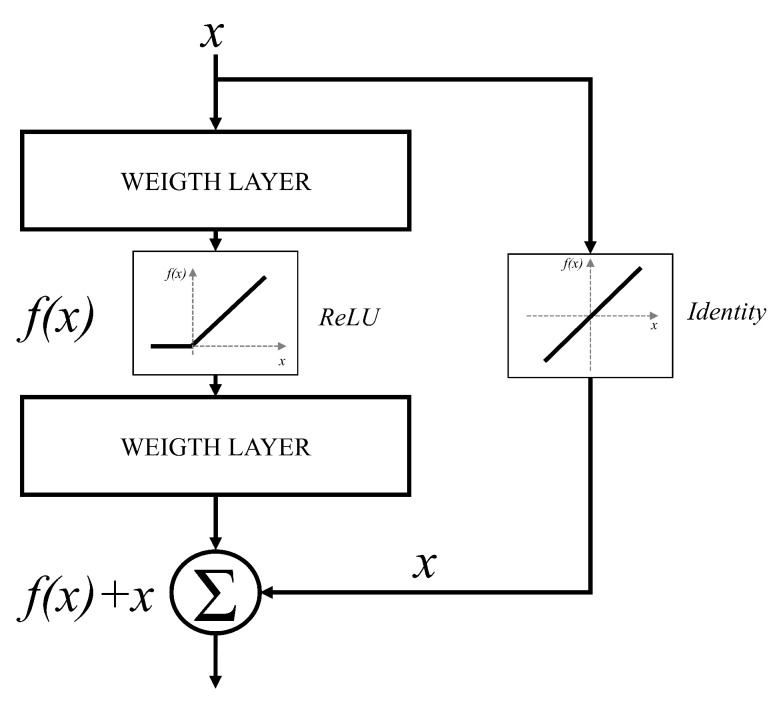
Schematic overview of a residual block.

**Figure 6 jpm-11-00028-f006:**
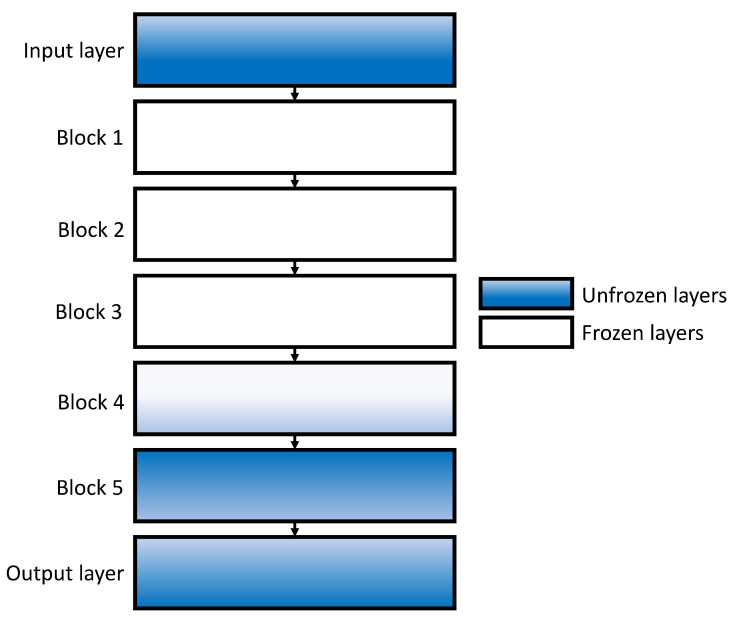
A schematic representation of freezing methodology on ResNet architecture.

**Figure 7 jpm-11-00028-f007:**
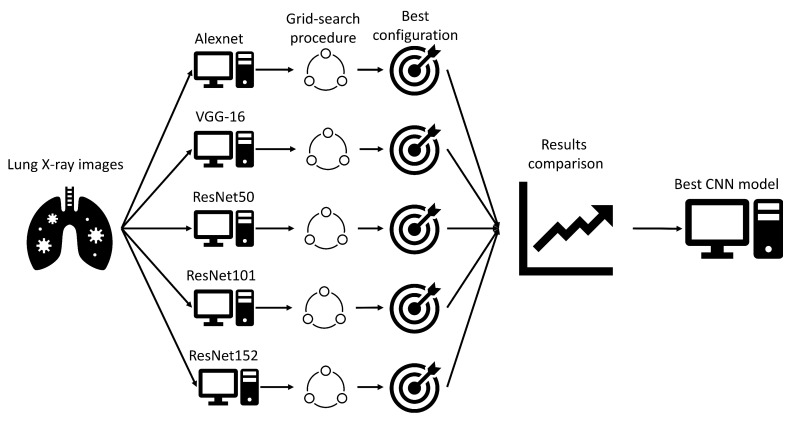
A schematic representation of the presented research methodology.

**Figure 8 jpm-11-00028-f008:**
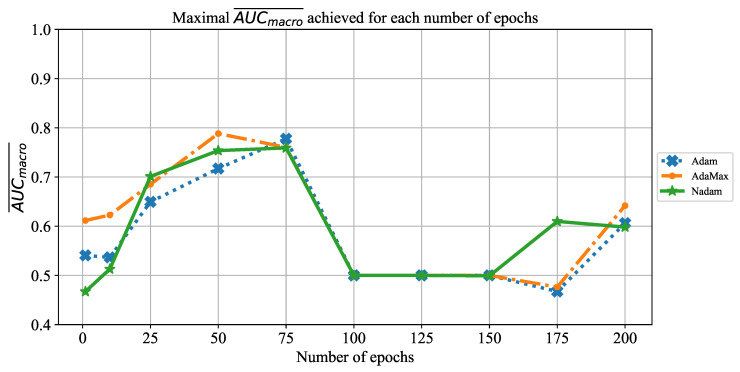
The change of maximal $\bar{AUC_{macro}}$ in dependence of the number of epochs for AlexNet.

**Figure 9 jpm-11-00028-f009:**
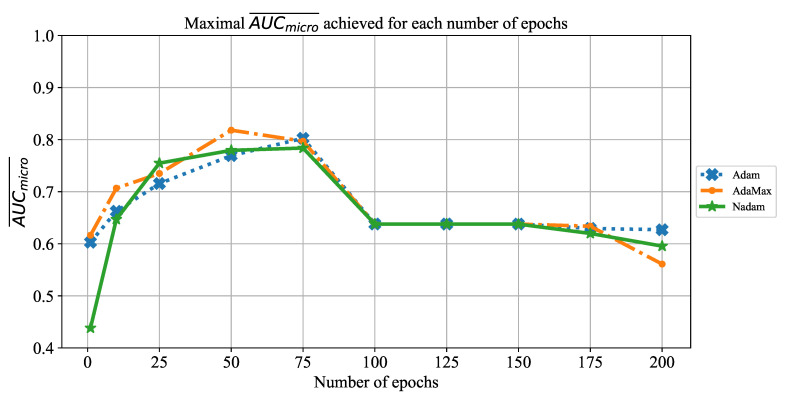
The change of maximal $\bar{AUC_{micro}}$ in dependence of the number of epochs for AlexNet.

**Figure 10 jpm-11-00028-f010:**
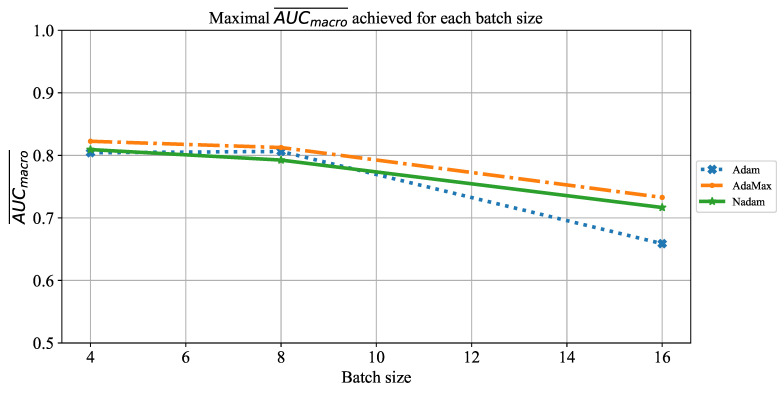
The change of maximal $\bar{AUC_{macro}}$ in dependence of batch size for AlexNet.

**Figure 11 jpm-11-00028-f011:**
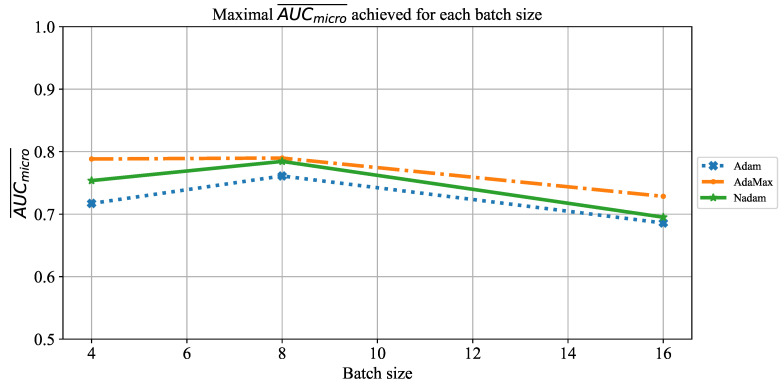
The change of maximal $\bar{AUC_{micro}}$ in dependence of batch size for AlexNet.

**Figure 12 jpm-11-00028-f012:**
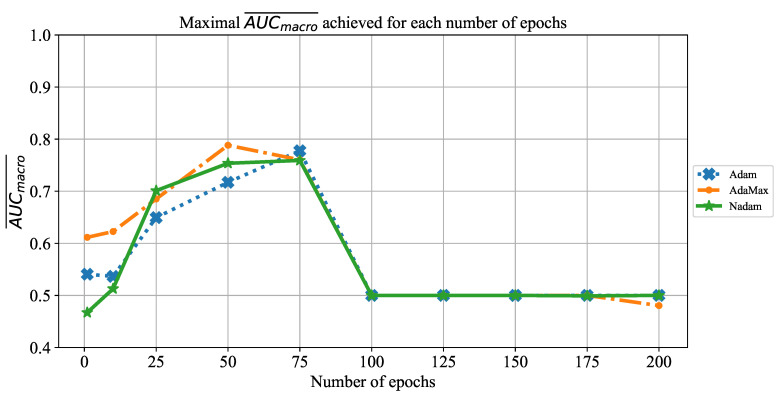
The change of maximal $\bar{AUC_{macro}}$ in dependence of number of epochs for VGG-16.

**Figure 13 jpm-11-00028-f013:**
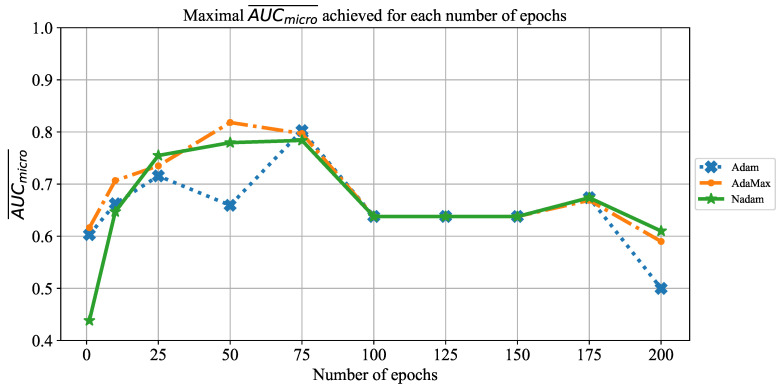
The change of maximal $\bar{AUC_{micro}}$ in dependence of number of epochs for VGG-16.

**Figure 14 jpm-11-00028-f014:**
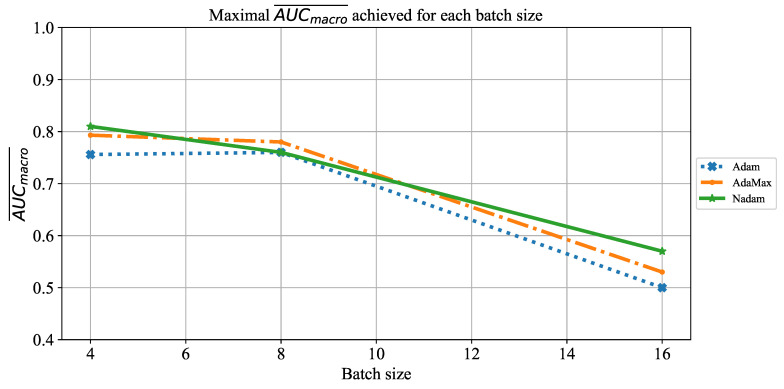
The change of maximal $\bar{AUC_{macro}}$ in dependence of batch size for VGG-16.

**Figure 15 jpm-11-00028-f015:**
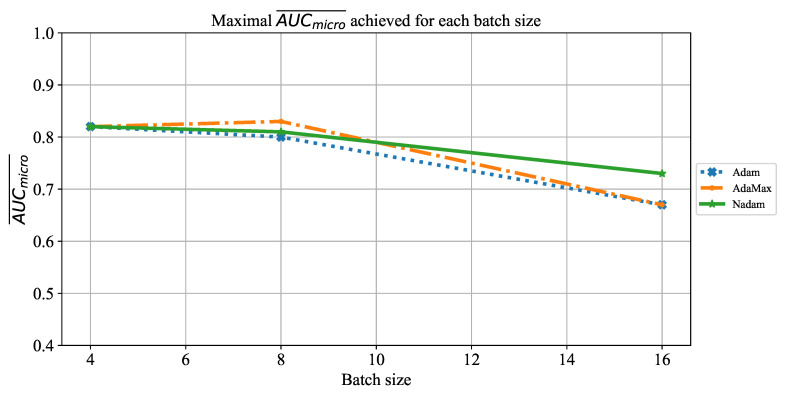
The change of maximal $\bar{AUC_{micro}}$ in dependence of batch size for VGG-16.

**Figure 16 jpm-11-00028-f016:**
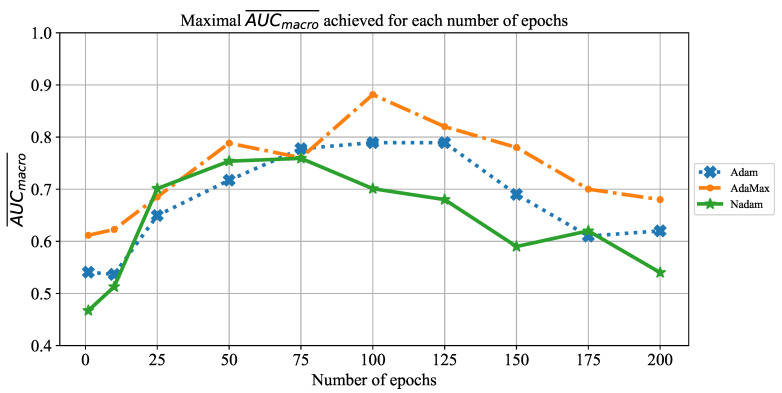
The change of maximal $\bar{AUC_{macro}}$ in dependence of number of epochs for ResNet50.

**Figure 17 jpm-11-00028-f017:**
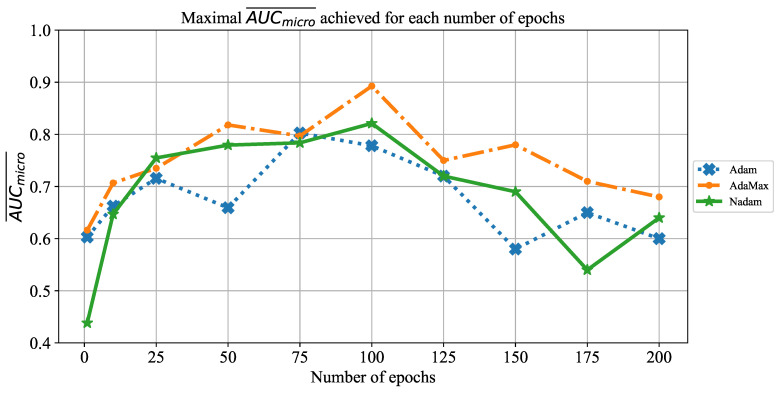
The change of maximal $\bar{AUC_{micro}}$ in dependence of the number of epochs for ResNet50.

**Figure 18 jpm-11-00028-f018:**
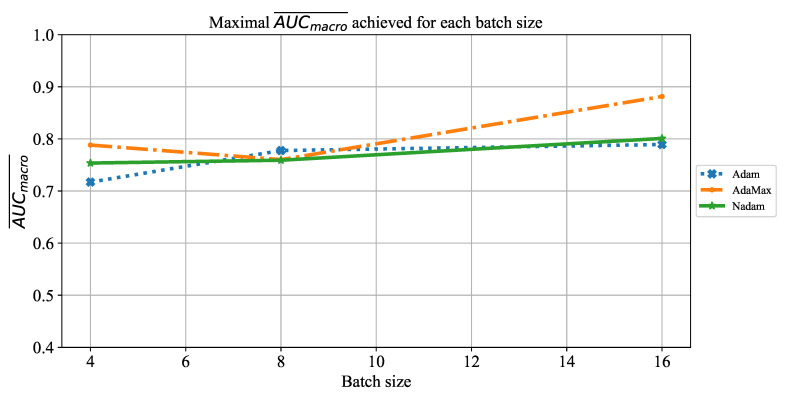
The change of maximal $\bar{AUC_{macro}}$ in dependence of batch size for ResNet50.

**Figure 19 jpm-11-00028-f019:**
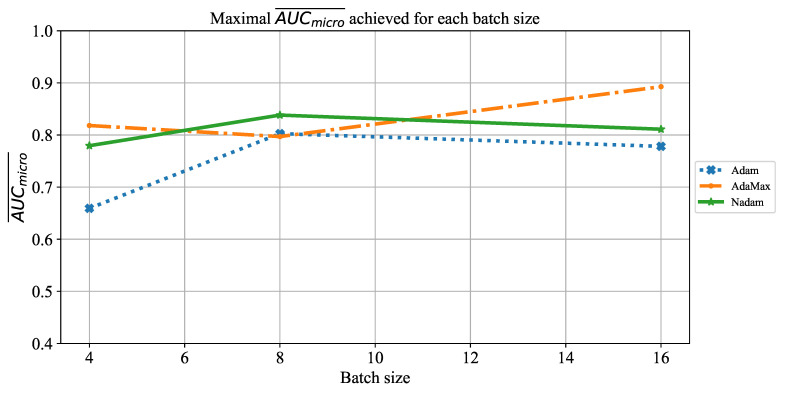
The change of maximal $\bar{AUC_{micro}}$ in dependence of batch size for ResNet50.

**Figure 20 jpm-11-00028-f020:**
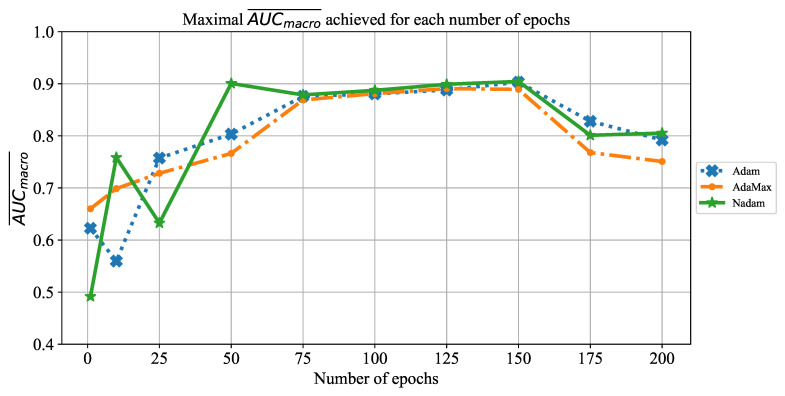
The change of maximal $\bar{AUC_{macro}}$ in dependence of number of epochs for ResNet101.

**Figure 21 jpm-11-00028-f021:**
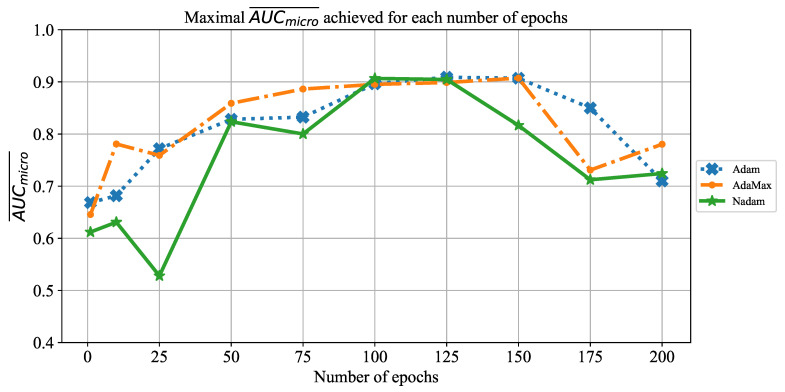
The change of maximal $\bar{AUC_{micro}}$ in dependence of number of epochs for ResNet101.

**Figure 22 jpm-11-00028-f022:**
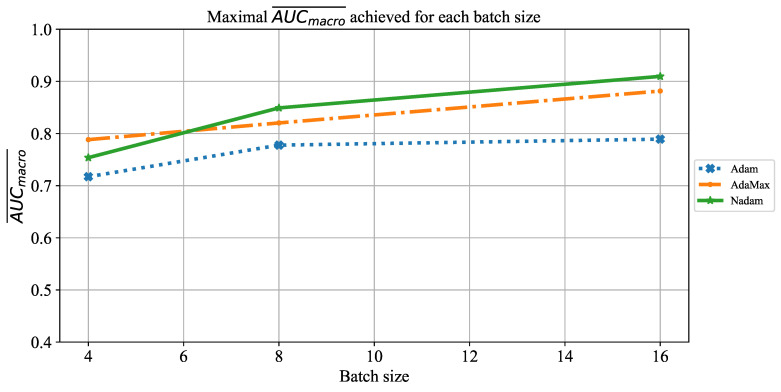
The change of maximal $\bar{AUC_{macro}}$ in dependence of batch size for ResNet101.

**Figure 23 jpm-11-00028-f023:**
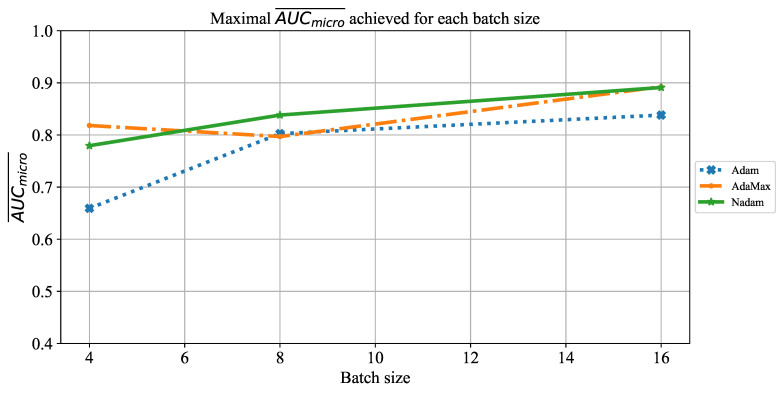
The change of maximal $\bar{AUC_{micro}}$ in dependence of batch size for ResNet101.

**Figure 24 jpm-11-00028-f024:**
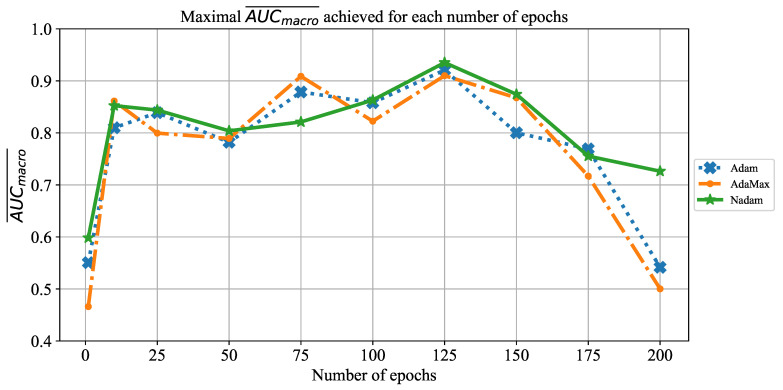
The change of maximal $\bar{AUC_{macro}}$ in dependence of number of epochs for ResNet152.

**Figure 25 jpm-11-00028-f025:**
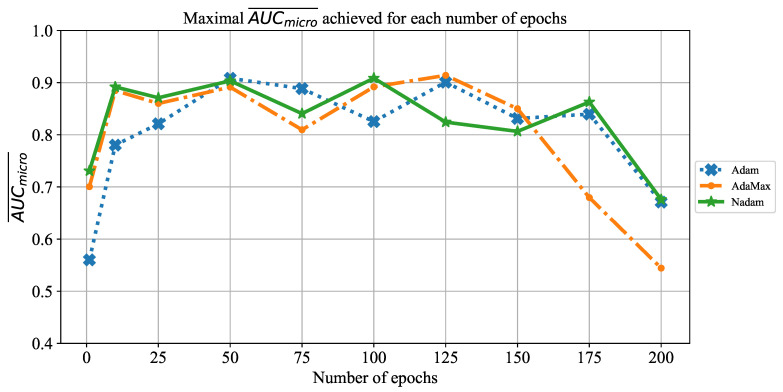
The change of maximal $\bar{AUC_{micro}}$ in dependence of number of epochs for ResNet152.

**Figure 26 jpm-11-00028-f026:**
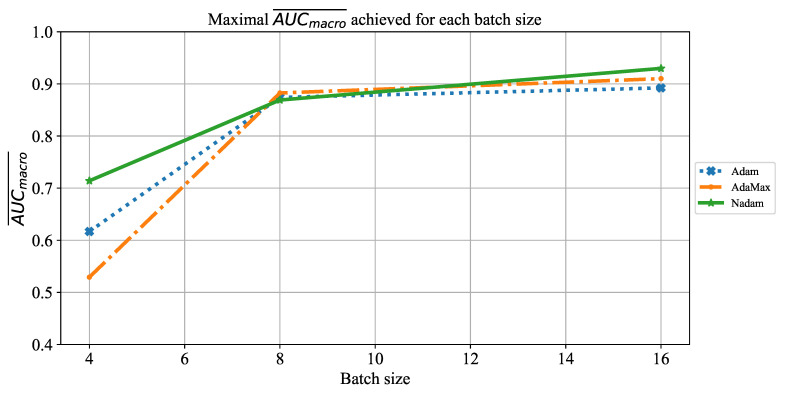
The change of maximal $\bar{AUC_{macro}}$ in dependence of batch size for ResNet152.

**Figure 27 jpm-11-00028-f027:**
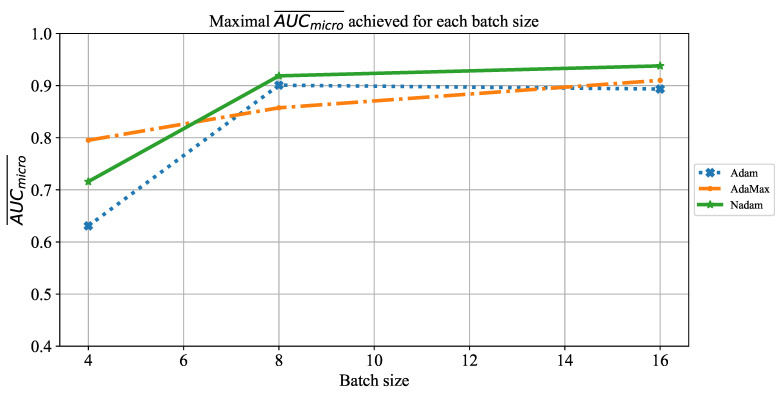
The change of maximal $\bar{AUC_{micro}}$ in dependence of batch size for ResNet152.

**Figure 28 jpm-11-00028-f028:**
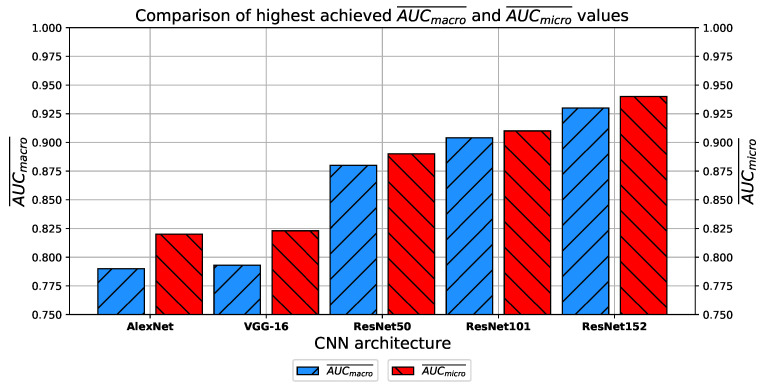
Comparison of highest $\bar{AUC_{macro}}$ and $\bar{AUC_{micro}}$ achieved with every CNN architecture.

**Figure 29 jpm-11-00028-f029:**
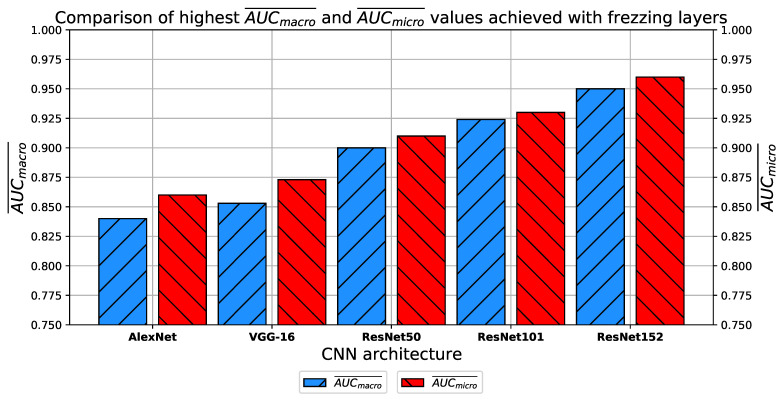
Comparison of highest $\bar{AUC_{macro}}$ and $\bar{AUC_{micro}}$ achieved with every CNN architecture and freezing layers.

**Table 1 jpm-11-00028-t001:** An overview of CNN hyper-parameters used during the grid-search procedure.

Number of Epochs	Solver	Batch Size
1	Adam [56]	2
5	Adamax [57]	4
10	Nadam [58]	8
25	-	16
50	-	-
75	-	-
100	-	-
125	-	-
150	-	-
175	-	-
200	-	-

**Table 2 jpm-11-00028-t002:** Parameters of each optimizer used in this research.

Solver	η	$\beta_{1}$	$\beta_{2}$	ϵ
Adam	0.001	0.9	0.99	$1\times 10^{-8}$
Adamax	0.02	0.9	0.999	$1\times 10^{-7}$
Nadam	0.001	0.9	0.999	$1\times 10^{-7}$

**Table 3 jpm-11-00028-t003:** Description of AlexNet architecture (C—convolutional layer, P—Max pooling, FC—fully connected).

Layer	Type	Feature Map	Size	Kernel Size	Stride	Activation Function
Input	Image	1	227×227×1	-	-	-
1	C	96	55×55×96	11×11	4	ReLU
	P	96	27×27×96	3×3	2	-
2	C	256	27×27×256	5×5	1	ReLU
	P	256	13×13×256	3×3	2	-
3	C	384	13×13×384	3×3	1	ReLU
4	C	384	13×13×384	3×3	1	ReLU
5	C	256	13×13×256	3×3	1	ReLU
	P	256	6×6×256	3×3	2	-
6	FC	-	9216	-	-	ReLU
7	FC	-	4096	-	-	ReLU
8	FC	-	4096	-	-	ReLU
Output	FC	-	4	-	-	Softmax

**Table 4 jpm-11-00028-t004:** Description of VGG 16 architecture (C—convolutional layer, P—Max pooling, FC—fully connected).

Layer	Type	Feature Map	Size	Kernel Size	Stride	Activation Function
Input	Image	1	224×224×1	-	-	-
1	2×C	96	224×224×64	3×3	1	ReLU
	P	64	112×112×64	3×3	2	-
3	2×C	128	112×112×128	3×3	1	ReLU
	P	256	56×56×128	3×3	2	-
5	2×C	256	56×56×256	3×3	1	ReLU
	P	384	28×28×256	3×3	2	ReLU
7	3×C	512	28×28×512	3×3	1	ReLU
	P	256	14×14×512	3×3	2	-
10	3×C	512	14×14×512	3×3	1	ReLU
	P	512	7×7×512	3×3	2	-
13	FC	-	25,088	-	-	ReLU
14	FC	-	4096	-	-	ReLU
15	FC	-	4096	-	-	ReLU
Output	FC	-	4	-	-	Softmax

**Table 5 jpm-11-00028-t005:** Overview of configurations that achieved highest $\bar{AUC_{macro}}$ for all CNN architectures.

Network	Number of Epochs	Batch Size	Solver
AlexNet	50	4	AdaMax
VGG-16	50	4	AdaMax
ResNet50	100	16	AdaMax
ResNet101	50	16	Nadam
ResNet152	125	16	Nadam

**Table 6 jpm-11-00028-t006:** Overview of configurations that achieved highest $\bar{AUC_{micro}}$ for all CNN architectures.

Network	Number of Epochs	Batch Size	Solver
AlexNet	50	4	AdaMax
VGG-16	50	4	AdaMax
ResNet50	100	16	AdaMax
ResNet101	100	16	Nadam
ResNet152	100	16	Nadam

**Table 7 jpm-11-00028-t007:** Representation of distribution of frozen and unfrozen layers with classification performances.

Network	Frozen Layers	Unfrozen Layers
AlexNet	1–5	6–9
VGG-16	1–12	13–16
ResNet50	1–42	43–50
ResNet101	1–92	93–101
ResNet152	1–139	140–152

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
