# Peer review of "Automatic Evaluation of the Lung Condition of COVID-19 Patients Using X-ray Images and Convolutional Neural Networks"

_jpm, 2021, doi:10.3390/jpm11010028_

Round 1

Reviewer 1 Report

In the reviewed paper, the authors proposed lung condition evaluation using CNNs. In general, the presented idea is interesting, but some issues should be improved like:
1) Add more discussion about the Internet of Medical Things, where your solution can be applied.
2) Why did you choose ResNet152?
3) Add a mathematical model of the training algorithm.
4) What about freezing layers?
5) Some comparisons with other solutions from the last 3 years should be added and extended.
6) Add comparison with other learning transfer models and freezing layers.

Reviewer 2 Report

The study is well structured and presents a well defined problem. The proposed methodology is not novel, however the tackling of coronavirus diagnosis is. The flow of the paper and its academic writing is making it an easy read. Some textual and more general comments:

  • IMO the last two paragraphs of Introduction should be in reversed order.
  • L73-74 state that the study deals with multiclass classification unlike other articles, however L64 states the results of a multiclass classification study (even if it is lower). Please rephrase.
  • L89 maybe a reference (a webpage?) for the Klinical Cetrer (Klinical or Clinical?, if it is the name keep the K, if the term clinical please change).
  • L110-111 needs rephrasing first sentence is too weak and second sentence uses "presented" in a back-to-back fashion
  • In Figure 2 percentages would be useful
  • L113 correction "a procedure", L114 "The augmentation procedure" (perhaps "process")
  • L134 even though I agree with not using scaling as an augmentation method, I have objections on whether other geometrical and brightness related augmentations could not be useful.
  • L137 I don't understand what "in a presented way" means
  • L146 which aforementioned CNN?
  • L199 "On the other hand, FPR can be defined as a ration of the number of incorrect" change "ration" to "ratio" (this mistake in recurring in the text)
  • L227-228: where can be noticed?? there is no results yet. First present the results, later describe what can be noticed
  • 5. Conclusion (change to 5. Conclusions)

Why was Adam, AdaMax and RMSprop(especially this) used in this study? There are more especially for multiclass classifications.

At no point of the text, overfitting was mentioned. Such complex architectures tend to overfit especially with limited data as in this study. did check for overfitting and if yes, how did you take care of it in this study?

Figures 7-14 show that the AUC drops while the iterations increase. How do you explain that? Is it expected? And why it is not the case in ResNet?

The discussion and conclusions should be a bit expanded a bit more. My serious concern is the overfitting in the study which it appears it wasn't taken care of at all. How did the number of iterations was determined? Was there a reason that there where these numbers or it was just a random comparison?

Round 2

Reviewer 1 Report

It can be accepted in the current form.

Author Response

The authors want to thank Reviewer 1 for constructive comments that have greatly improved this article and the research itself.

Reviewer 2 Report

The paper has been improved drastically. Only one comment on L127 and L130 where the paragraphs should be merged. Other than that, please check for minor spelling and syntax mistakes, if there are some and proceed.

Congratulations on your work and especially your adaptability to address the comments so fast and efficiently.

Author Response

The authors want to thank Reviewer 2 for constructive comments that have greatly improved this article and the research itself. All changes to the manuscript that were made according to Reviewers comments are marked with blue color.

The paper has been improved drastically. Only one comment on L127 and L130 where the paragraphs should be merged.

According to the comment, two paragraphs are merged forming:

“To summarize the novelty, this article, unlike the articles presented in the literature review, deals with a multi-class classification of X-ray images of positive COVID-19 patients with the aim of estimating the clinical picture. All the examples have used large number of images (larger than 1000) in the training and testing processes of the neural network. While the number of the COVID-19 patients is high, data collection, especially in countries with lower healthcare system quality, may be problematic due to the strain exhibited by the coronavirus. Because of this, it is important to test the possibility of algorithm development combined with data augmentation operations, which is the secondary goal of the presented research.”

Other than that, please check for minor spelling and syntax mistakes, if there are some and proceed.

Spelling and syntax mistakes are corrected.

Congratulations on your work and especially your adaptability to address the comments so fast and efficiently.

The authors would like to thank the reviewer for the comments and for the constructive suggestions that greatly helped to increase the quality of the article.